# Preventive and curative dental services utilization among children aged 12 years and younger in Tehran, Iran, based on the Andersen behavioral model: A generalized structural equation modeling

Elaheh Amirian[1,2], Samaneh Razeghi[1,2], Alireza Molaei[3], Ahmad R. Shamshiri[1,2], Simin Z. Mohebbi[1,2]*

1 Dentistry Research Institute, Research Center for Caries Prevention, Tehran University of Medical Sciences, Tehran, Iran, 2 Department of Community Oral Health, School of Dentistry, Tehran University of Medical Sciences, Tehran, Iran, 3 Department of Epidemiology and Biostatistics, School of Public Health, Tehran University of Medical Sciences, Tehran, Iran

* smohebbi@tums.ac.ir

**Data Availability Statement:** All relevant data are within the paper and its Supporting Information.

## Abstract

World Health Organization invites the nations to progress towards universal health care coverage. This study evaluated preventive and curative dental services utilization among children aged 12 years and younger in Tehran, Iran, based on the Andersen behavioral model using a generalized structural equation modeling. A phone-based cross-sectional study was conducted in Tehran, Iran, on 886 children in 2023. Information on curative and preventive/consultation dental service utilization and associated factors was collected by a standard questionnaire. We used a generalized structural equation model (GSEM) to build a model based on Andersen's behavioral model. Of 886 children, 22.1% used curative dental services, and 19.9% used preventive/consultation services in the past year. Among children aged 6 years and younger, age (OR = 1.87, p-value <0.001) and parent-perceived oral health need (OR = 54.77, p-value <0.001) predicted curative services utilization and age (OR = 1.45, p-value <0.001), knowledge (OR = 1.36, p-value <0.001), dental visit before the age of one (OR = 6.05, p-value = 0.04), and socioeconomic status (OR = 1.65, p-value = 0.03) predicted preventive/consultation services utilization. Regarding children aged 7 to 12, knowledge (OR = 1.28, p-value = 0.03), dental visit before the age of one (OR = 11.12, p-value = 0.02), socioeconomic status (OR = 2.53, p-value = 0.01), dental insurance (OR = 4.17, p-value <0.001), and parent-perceived oral health need (OR = 19.48, p-value <0.001) associated with curative dental services utilization, and dental visit before the age of one (OR = 10.05, p-value = 0.02), oral health behavior (OR = 1.25, p-value = 0.04), socioeconomic status (OR = 3.74, p-value <0.001), and parent-perceived oral health need (OR = 4.62, p-value <0.001) related to preventive/consultation services utilization. The findings of this study underscore the significant influence of predisposing, enabling, and need factors on dental services utilization among children aged 12 years and younger. These results

**Funding:** The present publication is funded by Research Center for Caries Prevention, Dentistry Research Institute, Tehran University of Medical Sciences (Grant number: 1401-2-238-56146). https://dri.tums.ac.ir. The funder had no role in study design, data collection and analysis, decision to publish or preparation of the manuscript.

**Competing interests:** he authors have declared that no competing interests exist.

provide a valuable perspective for policymakers, highlighting the modifiable determinants that could be targeted to improve dental services utilization in this age group.

## Introduction

The sustainable development agenda, adopted by all United Nations member states in 2015, outlines a collective plan to achieve peace and prosperity for everyone, both now and in the future. At its core are the 17 Sustainable Development Goals (SDGs) and 169 targets, which serve as urgent calls to action for both developed and developing countries. They recognize that ending poverty and other deprivations must go mutually with strategies that improve health and education, reduce inequality, and spur economic growth while preserving the environment. Ensuring the use of essential health services in the context of universal health coverage (UHC) is a prominent target of the SDGs, particularly SDG 3, which the member states are committed to achieving by 2030. SDG3 emphasizes good health and well-being for all [1]. Oral health is an inseparable part of health, encompassing a large segment of society and influencing their quality of life and financial resources [2].

Oral diseases are preventable, and appropriate oral health behaviors, including regular dental visits, are crucial. Nevertheless, not all people use these services regularly. Neglect of dental services utilization is more common in children than older age groups. A study in Brazil showed that 11.7% of individuals under 19 had never had a dental visit [3]. An integrative review by Curi et al. emphasizes age as an important indicator of dental services utilization. These results may be related to the cumulative effect of caries and the lack of parents' knowledge about the importance of early dental visits [3]. Regular dental visits in children can reduce dental caries, alleviate dental treatment costs, assess proper growth processes (such as anomalies and tooth eruption), and improve oral health-related quality of life [4].

Previous literature suggests that no single factor stands out as the most significant barrier to oral health care utilization; instead, several socioeconomic, familial, and mental factors affect dental visits by children. A study in Al-Madinah, Saudi Arabia, showed that children from high-income families use dental services more than those from lower-income families [5]. Another study in Brazil reported that the utilization of dental services was associated with children's age, mothers' education, family income, and dental caries [6]. A study in Lebanon highlighted the importance of economic and familial factors in dental services utilization among children [7]. Xu M et al. indicated that need factors were crucial in dental services utilization, whereas income showed no significant association [8].

Research on dental service utilization is vital to preparing equitable, high-quality health services for all individuals, communities, and populations. Theoretical models to explain health service utilization are crucial for guiding health services exploration. Using theoretical models in empirical designs and concepts is vital for substantially improving empirical design and outcomes for researchers [9]. Andersen's behavior model of health services utilization is a well-known framework for assessing the multifactorial nature of health service use.

A recent scoping review suggests that Andersen's theoretical model pivotally contributed to developing lasting core constructs, such as sociodemographics, health behaviors, and health system factors, in existing theoretical models to clarify the utilization of health services [9]. This model, established in 1968 and modified several times over the years, suggests that health services utilization is related to three groups of factors: predisposing (demographic, social

structure, and health beliefs), enabling (personal/family and social), and need (perceived and evaluated) [10].

The primary objective of this study was to assess the utilization of preventive and curative dental services among children aged 12 years and younger in Tehran, Iran. We used the Andersen behavioral model and generalized structural equation modeling to achieve this goal. By doing so, we aimed to provide insights into the factors influencing dental services utilization among children, thereby contributing to proper decision-making and addressing future challenges for policymakers.

## Materials and methods

This study was part of a cross-sectional, population-based telephone survey conducted in Tehran, the capital of Iran, in 2023. A total of 886 children aged 12 years and younger (423 children aged six years and younger and 463 children aged 7 to 12) were included in this study. Proportionate stratified random sampling was conducted to achieve a representative sample of children aged 12 and younger in 22 Tehran districts (Strata).

Based on the results of two pilot studies (face-to-face and telephone-based), we opted for telephone-based sampling as it offers a quick, low-cost representative sampling with a comfortable environment for participants [11].

It was necessary to ensure that the sample size was adequate for conducting a structural equation model, as recommended by Ramlall, who suggests a minimum of twenty observations per observed variable [12]. In our study, with 14 observed variables based on the theoretical model (Fig 1), this requirement translated to a minimum sample size of 280, which we comfortably exceeded with our sample of 886 children.

### Eligibility criteria

All children 12 years and younger who lived in Tehran and whose parents had access to a landline or mobile telephone and accepted participation in the study were included. Children whose parents were unable to answer questions due to hearing impairment or mental disorders were excluded from this program.

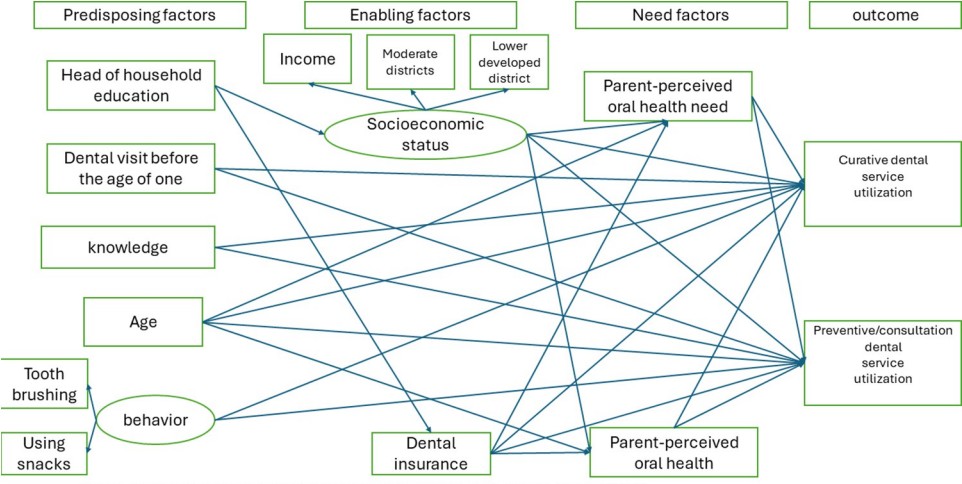

Figure 1. The theoretical model of study based on Andersen behavioral model of dental service utilization

**Fig 1. The theoretical model of study based on Andersen's behavioral model of dental service utilization.**

## Sample selection and data collection

The data collection was conducted between 1st Nov 2022 and 3rd Apr 2023. For recruitment, a list of phone numbers (75 numbers for each district), including landlines and cell phones, was generated using a random number-generating method, considering the area code of the desired district. Interviewers randomly selected phone numbers from this generated list. Twelve trained interviewers conducted calls to participants, where questions were posed to the parents. All children 12 years old and younger possessing eligibility criteria were included in each family. The process continued until the sample size for each district was achieved. Two phone calls were scheduled, the first in the morning and the second one was considered if the first one was not answered during the non-working hours in the evening. Each interview took 15 to 20 minutes.

Before starting the interviews, a two-hour orientation meeting was conducted to train the interviewers and explain the project objectives. Two monitored interviews were conducted. Project managers provided feedback to interviewers, answered their questions, and calibrated them. During data collection, an informed supervisor was present to check the accuracy and quality of the process. Additionally, the interviewers were provided with a telephone number so the project manager could resolve any issues during the data collection.

**Questionnaire.**   The tool used in this study was a comprehensive questionnaire designed by reviewing nationally and internationally approved questionnaires [4,13–16]. An expert panel of 13 professors specializing in community oral health, pediatric dentistry, public health, and epidemiology assessed the validity of the questionnaire. The expert panel selected the most relevant items to ensure clarity of the questions, interpretability, and accuracy across the questionnaire domains. This panel also evaluated the content validity of the items, assessing their relevance, coverage, and representativeness. The quantitative assessment involved measuring the "Content Validity Ratio" (CVR) and "Content Validity Index" (CVI). Modifications were made to address any contentious items until a consensus was reached. Furthermore, the questionnaire was piloted with 20 individuals from the target population outside the study sample. The questionnaire was again administered to the same 20 individuals using the test-retest method to assess reliability, achieving an actual agreement of more than 90%.

The questionnaire in this study followed Andersen's model components (Fig 1). The primary outcome of interest was dental care utilization, assessed through two questions: "Did your child have a dental visit in the past year?" Those who answered "yes" were further queried with: "Which type of dental services did your child receive? (curative or preventive/consultation)". Finally, dental service utilization was categorized into three groups: no utilization, curative services utilization, and preventive/consultation services utilization.

Specific measures were selected for inclusion as predictor variables, aligning with components from the Andersen model including a) predisposing factors such as age, gender, parents' oral health knowledge, head of household education, oral health behaviors (tooth brushing, snack consumption), and dental visits before age one; b) enabling factors encompassing socio-economic variables (monthly income, residential district), basic insurance, and dental insurance; and c) patient need items including parent-perceived oral health and perceived oral health needs (S1 File).

## Data handling and statistical analysis

The SPSS software, version 21, was utilized to summarize sample characteristics. Mean and standard deviation (SD) were used for continuous variables, while frequencies were used for categorical variables. The parents' oral health knowledge score was computed as the total sum

of scores obtained from seven questions. Correct answers were scored as one, and incorrect (including "I do not know") were scored as zero.

Other covariates included head of household education (categorized as less than diploma, diploma, associate and bachelor, master and more), basic insurance (yes or no), dental insurance (yes or no), parent-perceived oral health needs in the past year (yes or no), and parent-perceived oral health (categorized as very poor, poor, moderate, good, very good, excellent).

Missing data were generally low, at most 5% for any variables except income and dental insurance. For these variables, missing data were imputed using the Expectation-Maximization (EM) algorithm in SPSS software version 21. Equation-wise deletion was applied to handle missing data for other variables based on the structural equation modeling (SEM) rules.

Post-stratification survey weighting adjustments were conducted to mitigate inherent biases in the survey design and minimize the impact of challenges during data collection, such as overrepresented or underrepresented demographic groups, which could influence the results. Weights were calculated and applied to specific demographic variables, including the age and sex of the general population of children aged 12 years and younger in Tehran (S2 File).

## Generalized structural equation modeling

The study treated oral health behavior and socioeconomic status as latent constructs. Oral health behavior was measured through two questions: 1) "How often does your child brush their teeth?" (options: irregularly, once a day, more than once a day) and 2) "How often does your child consume snacks and sweet beverages?" (options: three times a day or more, once or twice a day, every week, every month or less). The history of dental visits before age one was considered an oral health behavior component. Still, it was treated separately due to its retrospective nature and added to the model as a distinct observed variable.

Two variables were used to measure socioeconomic status. First, household income per month was categorized into four groups: very poor (100 USD or less), poor (100–200 USD), moderate (200–300 USD), and rich/very rich (more than 300 USD). Second, the residence district in Tehran, which has 22 districts, was categorized into four strata based on a previous study that ranked the districts in terms of development and quality of life [17]. Affluent districts include districts 1–3, 6, and 22; moderate districts include 4, 5, 8, 13, 20, and 21; lower affluent districts include districts 7, 9, 11,12, 14–16, and19; and need intervention districts include districts 10, 17 and 18. We merged the two last groups and used the district variable as a trichotomous variable in the analysis (Affluent, moderate, and lower effluent).

The STATA software, version 17, was used for model construction. Variables with a p-value <0.2 in the bivariate analysis were considered. Given our multilevel categorical outcome variable, we utilized the generalized structural equation model (GSEM) to construct a model based on the three categories of variables in the Andersen behavioral model. GSEM, a type of structural equation modeling (SEM), accommodates both categorical and continuous outcomes and allows for any combination of observed variables in the model [18].

We utilized various model specifications, including logit and ordinal links and families such as Bernoulli, multinomial, and Gaussian, within the generalized structural equation model (GSEM). The model was separately applied to two age groups: children aged six years and younger and children aged 7 to 12 years, allowing for comparison between these groups.

Fig 1 illustrates our theoretical model, depicting the direct and indirect effects of predisposing, enabling, and need factors on dental services utilization. Receiver operating characteristic (ROC) analysis was employed to evaluate the model's predictive ability. All analyses were performed using the STATA software, version 17, with significance set at a p-value <0.05.

### Ethical considerations

At the outset, the interviewers introduced themselves, clarified the research objectives, and assured participants were interested in being involved in the study, emphasizing that names did not need to be disclosed and that the interviews were conducted in a confidential environment. As our study was a phone survey, we received verbal consent from the parents by asking whether they were willing to participate in this study. Their response was recorded via phone call. If someone is not eager to participate, they will not collaborate. Participants were informed of their right to cease participation or withdraw from the study at any time. In cases of technical issues or participant preference to terminate the interview prematurely, replacements were sought from the same age group and residential area. To ensure the quality of data collection, a quality control team reviewed ten percent of randomly selected interviews through recorded phone calls.

Ethical clearance was received from the Ethics Committee of Tehran University of Medical Science (IR.TUMS.DENTISTRY.REC.1401.094).

## Results

### Description of the study population

To achieve the sample size, 16258 calls were conducted, 5428 calls were answered, and 1322 calls led to the completion of the questionnaires.

The sample consisted of 886 children aged 12 years old and younger, with 423 children aged 6 and younger and 463 children aged 7–12 years old (mean = 6.71, SD = 3.32). Approximately 49.8% of the children were male (Table 1). About 19% of the children lived in affluent districts, 28% in moderate districts, and 53% in lower affluent districts. Most household heads had an educational level ranging from a high school diploma to a bachelor's degree (68.4%). Most children belonged to middle-income families (40.3%) (Table 1). Sixty-four percent of children had basic insurance, and 32.8% had dental insurance. According to parental perception, the majority of children (37.6%) had good oral health, and 66.8% did not report oral health needs in the past year (Table 1). The average score for parents' oral health knowledge was 4.41 ± 1.40 (ranging from 0 to 7).

Regarding oral health behaviors, 44.1% of children did not have a routine for tooth brushing, while 47.7% brushed their teeth once a day. Forty-four percent of children consumed snacks once or twice a day, and 25.3% consumed snacks three times a day or more. Additionally, 3.3% of children had a dental visit before age one (Table 1). Overall, 57.2% of children did not use dental services, 22.1% used curative services, and 19.9% used preventive/consultation services (Table 1).

### Predisposing factors and dental services utilization

Table 2 summarizes the frequency of dental services utilization based on predisposing, enabling, and need factors. The utilization rates for preventive/consultation and curative services were higher among children aged 7 to 12 compared to those aged six years and younger (20.7% versus 19.2% for preventive/consultation services and 29.9% versus 14% for curative services) (Table 2). The percentage of children who did not use dental services in the past year was notably higher among children from households with lower levels of education among the household heads (67.7% for heads of household with less than a diploma, 64.8% for those with a high school diploma, 49.7% for those with an associate or bachelor's degree, and 47.8% for those with a master's degree or higher) (Table 2). The pattern of dental service utilization had no significant difference in the two gender groups.

**Table 1. Predisposing, enabling, and need factors and dental service utilization in Tehran inhabitants 12 years old and younger children.**

| | | 6 years old and younger (n = 423) | 7 to 12 years old (n = 463) | 12 years old and less (n = 886) |
|---|---|---|---|---|
| **Predisposing factors:** | | | | |
| Gender | Male | 211 (49.9%) | 230 (49.5%) | 441 (49.8%) |
| | Female | 212 (50.1%) | 233 (50.5%) | 445 (50.2%) |
| Head of household education | Literate and less than diploma | 55 (13%) | 70 (15.1%) | 125 (14.2%) |
| | Diploma | 146 (34.5%) | 173 (37.4%) | 319 (36%) |
| | Associate and bachelor | 143 (33.8%) | 144 (31.1%) | 287 (32.4%) |
| | Master and more | 71 (16.8%) | 66 (14.3%) | 137 (15.5%) |
| Tooth brushing | Irregular | 217 (51.3%) | 174 (37.6%) | 391 (44.1%) |
| | Once a day | 178 (42.1%) | 245 (52.9%) | 423 (47.7%) |
| | More than once a day | 24 (5.7%) | 43 (9.3%) | 67 (7.6%) |
| Using snacks | Three times a day or more | 113 (26.7%) | 111 (24%) | 224 (25.3%) |
| | Once or twice a day | 167 (20.3%) | 223 (48.2%) | 390 (44%) |
| | Every week | 86 (13.9%) | 86 (18.6%) | 172 (19.4%) |
| | Every month | 49 (11.6%) | 43 (9.3%) | 92 (10.4%) |
| Dental visit before the age of one | No | 402(95%) | 438 (94.6%) | 840 (94.8%) |
| | Yes | 17 (4%) | 12 (2.6%) | 29 (3.3%) |
| **Enabling factors:** | | | | |
| Districts | Affluent | 77 (18.2%) | 88 (19%) | 165 (18.6%) |
| | Moderate | 125 (29.5%) | 124 (26.8%) | 249 (28.1%) |
| | Lower affluent | 222 (52.4%) | 250 (54.1%) | 472 (53.3%) |
| Income | Very poor | 95 (22.5%) | 123 (26.6%) | 218 (24.6%) |
| | Poor | 112 (26.5%) | 130 (28.1%) | 242 (27.3%) |
| | Moderate | 179 (42.3%) | 178 (38.4%) | 357 (40.3%) |
| | Rich and very rich | 37 (8.7%) | 32 (6.9%) | 69 (7.8%) |
| Basic insurance | No | 69 (16.3%) | 67 (14.5%) | 136 (15.3%) |
| | Yes | 353 (83.5%) | 393 (84.9%) | 746 (64.2%) |
| Dental insurance | No | 249(58.9%) | 292 (63.1%) | 541 (61.1%) |
| | Yes | 142 (33.6%) | 149 (32.2%) | 291 (32.8%) |
| **Need factors** | | | | |
| Parent-perceived oral health | Very poor | 14 (3.3%) | 10 (2.2%) | 24 (2.7%) |
| | Poor | 23 (5.4%) | 48 (10.4%) | 71 (8%) |
| | Moderate | 71 (16.7%) | 90 (19.5%) | 161 (18.2%) |
| | Good | 145 (34.2%) | 188 (40.7%) | 333 (37.6%) |
| | Very good | 47 (11.1%) | 57 (12.3%) | 104 (11.7%) |
| | Excellent | 124 (29.2%) | 69 (14.9%) | 193 (21.8% |
| Parent-perceived need | No | 324 (76.6%) | 268 (57.9%) | 592 (66.8%) |
| | Yes | 92 (21.7%) | 192 (41.5%) | 284 (32.1%) |
| **Dental service utilization** | No | 280 (66.2%) | 227 (49%) | 507 (57.2%) |
| | Curative | 59 (13.9%) | 137 (29.6%) | 196 (22.1%) |
| | Preventive/consultation | 81 (19.1%) | 95 (20.5%) | 176 (19.9%) |

The data in Table 2 indicates that more regular tooth brushing correlates with a higher frequency of preventive/consultation services utilization (32.8% among those who brush their teeth more than once a day versus 12.9% among those who brush their teeth irregularly).

Additionally, half of the children who had their first dental visit before the age of one received preventive/consultation services in the past year (Table 2).

Table 2. Dental service utilization in Tehran inhabitant children aged 12 years old and less (n = 886) based on Andersen's behavioral model.

| Variables | | No utilization | Curative service utilization | Preventive/consultation service utilization |
|---|---|---|---|---|
| **Predisposing factors:** | | | | |
| Age | 6 years old and less | 281 (66.7%) | 59 (14%) | 81 (19.2%) |
| | 7 to 12 years old | 226 (49.3%) | 137 (29.9%) | 95 (20.7%) |
| Gender | Female | 251 (56.9% | 99 (22.4%) | 91 (20.6%) |
| | Male | 256 (58.4%) | 97 (22.1%) | 85 (19.4%) |
| Head of household education | Less than diploma | 84 (67.7%) | 28 (23.3%) | (23.3%) |
| | Diploma | 204 (64.8%) | 69 (21.9%) | 42 (13.3%) |
| | Associate and bachelor | 142 (49.7%) | 67 (23.4%) | 77 (26.9%) |
| | Master and more | 65 (47.8%) | 27 (19.9%) | 44 (32.4%) |
| Tooth brushing | Irregular | 262 (67.7%) | 75 (19.4%) | 50 (12.9%) |
| | Once a day | 210 (50%) | 108 (25.7%) | 102 (24.3%) |
| | More than once a day | 32 (47.8%) | 13 (19.4%) | 22 (32.8%) |
| Using snacks | Three times a day or more | 115 (52%) | 56 (25.3%) | \50 (22.6%) |
| | Once or twice a day | 214 (55.3%) | 90 (23.3%) | 83 (21.4%) |
| | Every week | 103 (60.2%) | 38 (22.2%) | 30 (17.5%) |
| | Every month or less | 68 (73.9%) | 12 (13%) | 12 (13%) |
| Dental visit before the age of one | No | 491 (58.9%) | 189 (22.7%) | 154 (18.5%) |
| | Yes | 8 (28.6%) | 6 (21.4%) | 14 (50%) |
| **Enabling factors:** | | | | |
| Districts | Affluent | 71 (43.6%) | 48 (29.4%) | 44 (27%) |
| | Moderate | 145 (58.5%) | 52 (21%) | 51 (20.6%) |
| | Lower affluent | 291 (62.2%) | 96 (20.5%) | 81 (17.3%) |
| Income | Very poor | 142 (65.7%) | 46 (21.3%) | 28 (13%) |
| | Poor | 152 (63.3%) | 56 (23.3%) | 32 (13.3%) |
| | Moderate | 185 (52.1%) | 75 (21.1%) | 95 (26.8%) |
| | Rich and very rich | 28 (41.2%) | 19 (27.9%) | 21 (30.9%) |
| Basic insurance | No | 82 (61.2%) | 26 (19.4%) | 26 (19.4%) |
| | Yes | 422 (57%) | 169 (22.8%) | 150 (20.2%) |
| Dental insurance | No | 330 (60.9%) | 117 (21.6%) | 95 (17.5%) |
| | Yes | 156 (50.6%) | 74 (24%) | 78 (25.3%) |
| **Need factors:** | | | | |
| Parent-perceived oral health | Very poor | 13 (56.5%) | 10 (43.5%) | 0 |
| | Poor | 37 (52.9%) | 26 (37.1%) | 7 (10%) |
| | Moderate | 78 (49.1%) | 51 (32.1%) | 30 (19.7%) |
| | Good | 204 (61.4) | 66 (19.9%) | 62 (18.7%) |
| | Very good | 57 (55.3%) | 22 (21.4%) | 24 (23.3%) |
| | Excellent | 118 (61.5% | 21 (10.9%) | 53 (27.6%) |
| Parent-perceived need | No | 413 (70.1%) | 47 (8%) | 129 (21.9%) |
| | Yes | 89 (31.8%) | 148 (52.9%) | 43 (15.4%) |

## Enabling factors and dental services utilization

No dental services utilization was more common in less affluent districts, with 62.2% of children falling into this category. Conversely, in affluent districts, the frequency of preventive/consultation services utilization was higher at 27%, and curative services utilization was even higher at 29.4% (Table 2).

Regarding family income, there was a noticeable decrease in the frequency of no utilization of dental services among higher income groups, with 41.2% reporting no utilization compared

to 65.7% in the lowest income group. Conversely, the trend was reversed for preventive/consultation services, with 30.9% utilization in the higher-income group versus 13% in the lowest-income group (Table 2).

### Need factors and dental services utilization

Most children whose parents perceived them to have oral health needs utilized curative services (52.9%). Conversely, among children without perceived oral health needs, the majority did not receive any dental services (70.1%).

Moreover, the utilization of preventive/consultation services was more prevalent among children from families where parents perceived their oral health as better (27.6% in better-off families compared to 0% in the poorest families) (Table 2).

### Generalized structural equation model results

Figs 2 and 3 show the final generalized structural equation model. Tables 3 and 4 represent the GSEM results according to the two age groups.

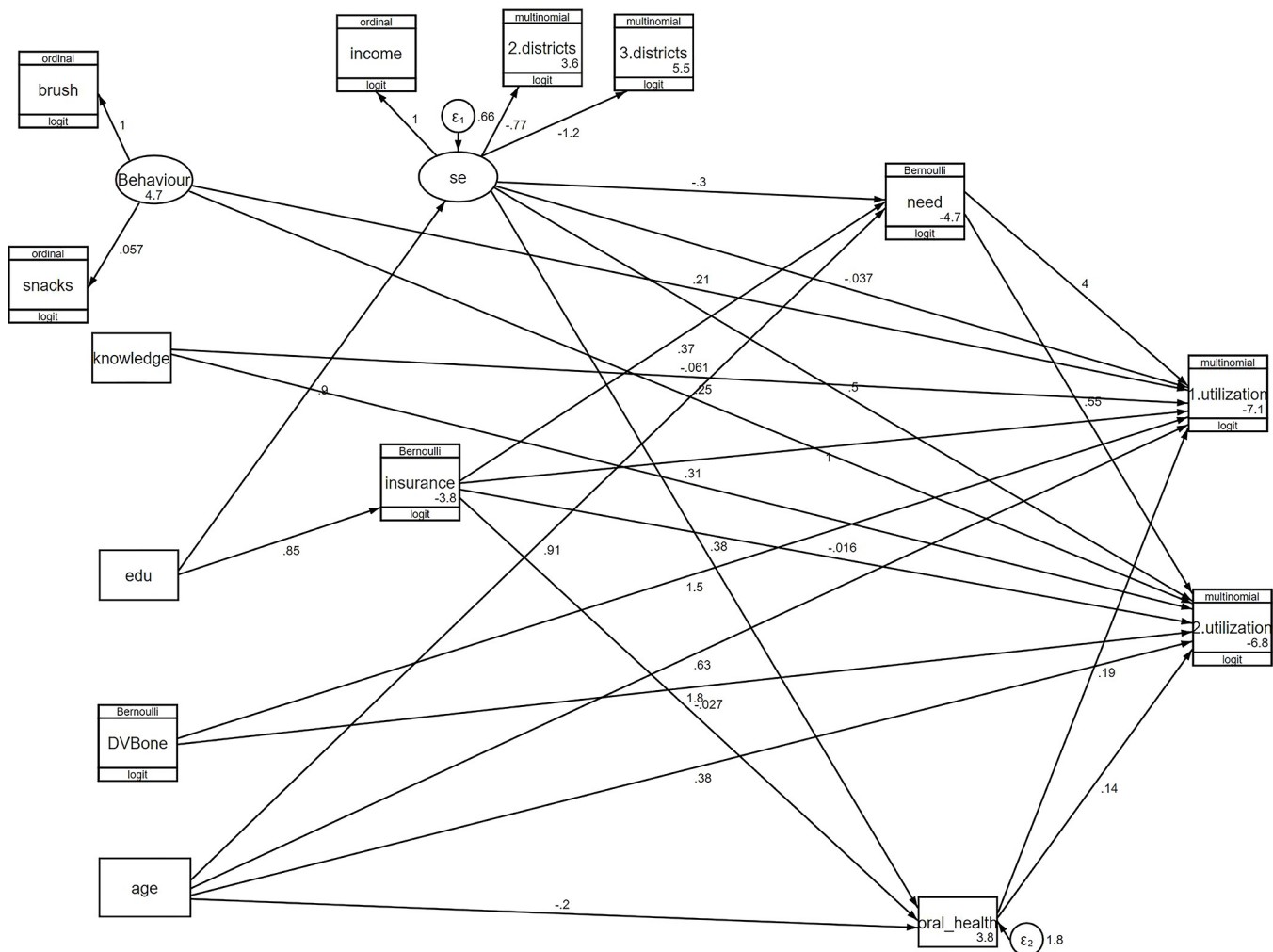

**Fig 2. Generalized structural equation model for Tehran inhabitant children aged six years and younger B) 7–12 year-old.**

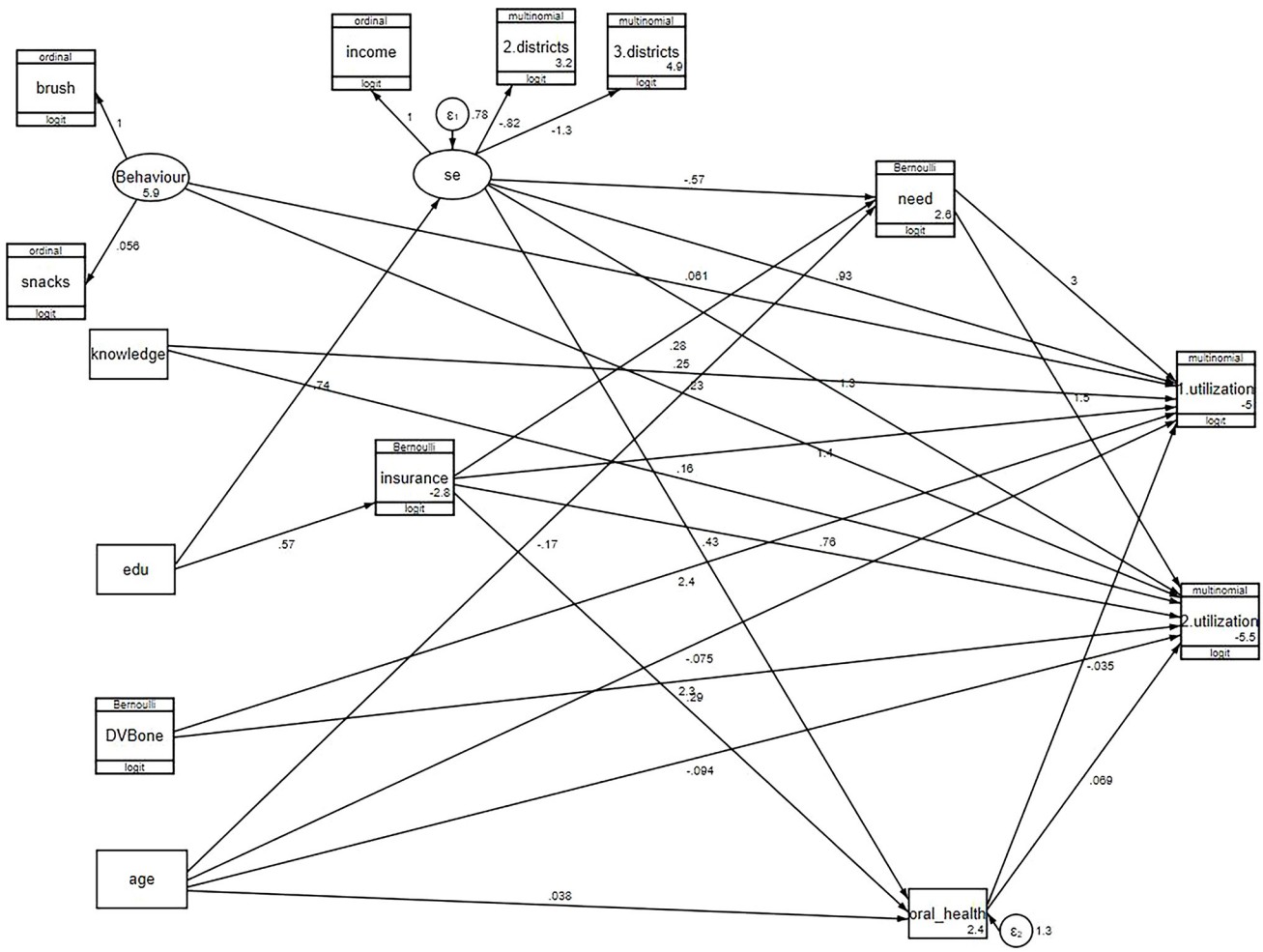

**Fig 3. Generalized structural equation model for Tehran inhabitant children aged 7–12 year-old.**

**Table 3. Modeling of curative dental service utilization in Tehran inhabitant 12 years old and younger children (n = 886) based on Andersen's behavioral model.**

| | Children aged 6 years and younger | | | Children aged 7–12 years | | |
|---|---|---|---|---|---|---|
| | Odds Ratio | Confidence interval | p-value | Odds Ratio | Confidence interval | p-value |
| **Predisposing factors:** | | | | | | |
| Child age | 1.87 | 1.34–2.61 | 0.00 | 0.92 | 0.77–1.11 | 0.43 |
| Knowledge | 0.94 | 0.63–1.38 | 0.75 | 1.28 | 1.01–1.63 | 0.03 |
| Dental visit before the age of one | 4.36 | 0.43–43.77 | 0.21 | 11.12 | 1.43–86.04 | 0.02 |
| Oral health behavior | 1.23 | 0.69–2.19 | 0.46 | 1.06 | 0.89–1.25 | 0.47 |
| **Enabling factors:** | | | | | | |
| Socioeconomic status | 0.96 | 0.45–2.03 | 0.92 | 2.53 | 1.18–5.40 | 0.01 |
| Dental insurance | 2.85 | 0.95–8.56 | 0.06 | 4.17 | 2.05–8.49 | 0.00 |
| **Need factors:** | | | | | | |
| Parent-perceived oral health needs | 54.77 | 16.64–180.20 | 0.00 | 19.48 | 7.62–49.78 | 0.00 |
| Parent-perceived oral health | 1.21 | 0.86–1.70 | 0.26 | 0.96 | 0.70–1.32 | 0.82 |

**Table 4.  Modeling of preventive/consultation dental service utilization in Tehran inhabitants 12 years old and younger children (n = 886) based on Andersen's behavioral model.**

| | Children aged 6 years and younger | | | Children aged 7–12 old | | |
|---|---|---|---|---|---|---|
| | Odds Ratio | Confidence interval | p-value | Odds Ratio | Confidence interval | p-value |
| **Predisposing factors:** | | | | | | |
| Child age | 1.45 | 1.19–1.78 | 0.00 | 0.91 | 0.73–1.12 | 0.37 |
| Knowledge | 1.36 | 1.08–1.70 | 0.007 | 1.17 | 0.92–1.49 | 0.17 |
| Dental vist before the age of one | 6.05 | 1.06–34.44 | 0.04 | 10.05 | 1.32–76.53 | 0.02 |
| Oral health behavior | 1.27 | 0.91–1.79 | 0.15 | 1.25 | 1.006–1.57 | 0.04 |
| **Enabling factors:** | | | | | | |
| Socioeconomic status | 1.65 | 1.03–2.63 | 0.03 | 3.74 | 1.44–9.720 | 0.00 |
| Dental insurance | 0.98 | 0.49–1.97 | 0.96 | 2.14 | 0.99–4.60 | 0.05 |
| **Need factors:** | | | | | | |
| Parent-perceived oral health needs | 1.73 | 0.57–5.28 | 0.33 | 4.62 | 1.72–12.41 | 0.00 |
| Parent-perceived oral health | 1.15 | 0.87–1.51 | 0.29 | 1.07 | 0.75–1.52 | 0.70 |

## Causal network of dental services utilization in children aged 6 years old and younger

Tooth brushing exhibited the strongest contribution to oral health behavior, with a robust coefficient of 1. The snack consumption showed a robust coefficient of 0.05, which was not statistically significant (p-value = 0.47).

Regarding the socioeconomic construct, living in less affluent districts had the most significant impact (robust coefficient = -1.24, p-value <0.001). Living in moderate districts also showed a statistically significant effect (robust coefficient = -0.77, p-value = 0.007). Income had a robust coefficient of 1, indicating its positive impact within the model.

Each year, an increase in age corresponded to 1.87 times higher odds of curative services utilization (p-value <0.001). Children who had a dental visit before age one showed 4.36 times higher odds of curative services utilization, but this association was not statistically significant. Children reporting oral health needs in the past year exhibited significantly higher odds of curative services utilization by 54.77 times (p-value <0.001). Having dental insurance was associated with 2.85 times higher odds of curative services utilization, although this association was not statistically significant (Table 3).

The odds of preventive/consultation services utilization increased by 1.45 times for each increase in age by one year (p-value <0.001) and 1.36 times for each one-unit increase in knowledge (p-value = 0.00). Additionally, having a dental visit before the age of one increased the odds of preventive/consultation services utilization by 6.05 times (p-value = 0.04). Better socioeconomic status was also associated with higher odds of preventive/consultation services utilization (OR = 1.65, p-value = 0.03) (Table 4).

The education level of the household head directly predicted dental insurance (OR = 2.34, p-value <0.001) and socioeconomic status (robust coefficient = 0.90, p-value <0.001). The age of the child was directly associated with parent-perceived oral health needs in the past year (OR = 2.49, p-value <0.001) and indirectly linked to parent-perceived oral health (OR = 0.81, p-value <0.001). Additionally, better socioeconomic status was significantly associated with better parent-perceived oral health (OR = 1.46, p-value = 0.00).

## Causal network of dental services utilization in 7–12-year-old children

In the behavior construct, brushing teeth had the most robust coefficient. None of the observed variables within the behavior construct had a significant effect. In the socioeconomic

construct, living in less affluent districts had the most substantial contribution (robust coefficient = -1.33, p-value <0.001). The robust coefficient was 1 for income and was -0.82 for living in moderate districts (p-value = 0.00).

The odds ratio of having a curative dental visit was 11.12 (p-value = 0.02) in children with a history of a dental visit before the age of one and 1.28 in those with a higher knowledge score (p-value = 0.03). Better socioeconomic status and having dental insurance was positively related to curative services utilization (OR = 2.53, p-value = 0.01 for socioeconomic status and OR = 4.17, p-value <0.001 for dental insurance). Parent-perceived oral health needs in the past year were associated with higher odds of curative services utilization (OR = 19.48, p-value <0.001) (Table 3).

A dental visit before the age of one led to an increase of 10.05 times in the odds of using preventive/consultation services (p-value = 0.02). Better oral health behavior was positively related to preventive/consultation dental services utilization (OR = 1.25, p-value = 0.04). Parent-perceived oral health needs in the last year had a statistically significant positive relationship with preventive/consultation services utilization (OR = 4.62, p-value <0.001) (Table 4).

The education level of the household head had a statistically significant positive association with dental insurance and socioeconomic status (robust coefficient = 0.74, p-value <0.001 for socioeconomic status and OR = 1.76, p-value <0.001 for dental insurance).

The child's age and socioeconomic status were indirectly linked to parent-perceived oral health needs (OR = 0.84, p-value = 0.01 for age and OR = 0.56, p-value = 0.02 for socioeconomic status); higher socioeconomic status was associated with better parent-perceived oral health (OR = 1.53, p-value <0.001).

Fig 4 shows the ROC curve related to the model predictability of curative and preventive/consultation services utilization in two age groups. The area under the ROC curve (AUC) was 0.98 for curative services utilization and 0.97 for preventive/consultation services utilization in children aged 6 years old and younger. The AUC was 0.79 and 0.83 for curative and preventive/consultation services utilization in children aged 7 to 12, respectively.

## Discussion

Ensuring all populations access necessary health services is a key component of the Sustainable Development Goals (SDGs). This study's findings highlight dental services utilization, defined as the use of dental services within a specific timeframe, as a complex and multifactorial phenomenon [19]. Using Andersen's behavioral model, we analyzed dental services utilization and its associated factors through a generalized structural equation model. Understanding these underlying factors can assist policymakers and health authorities in improving dental service utilization patterns within the population [8].

In children aged six years old and younger, age and parent-perceived oral health needs were factors related to curative services utilization, while age and socioeconomic status were associated with preventive/consultation services utilization. In children aged 7 to 12, dental visits before age one, parent-perceived oral health needs, and socioeconomic status were associated with both curative and preventive/consultation services utilization. Additionally, dental insurance was related to curative services utilization in children aged 7 to 12 years.

Among children aged six years old and younger, 66.2% had no dental visits, 13.9% utilized curative services, and 19.1% used preventive/consultation services in the last year. For children aged 7 to 12, 49% did not use dental services in the past year, 29.6% used curative services, and 20.5% used preventive/consultation services. A study conducted in China, an Asian developed country, showed that 45.2% of children aged 2 to 6 years old used dental services, with 24.3% utilizing preventive services [8]. The frequency of dental services utilization among children

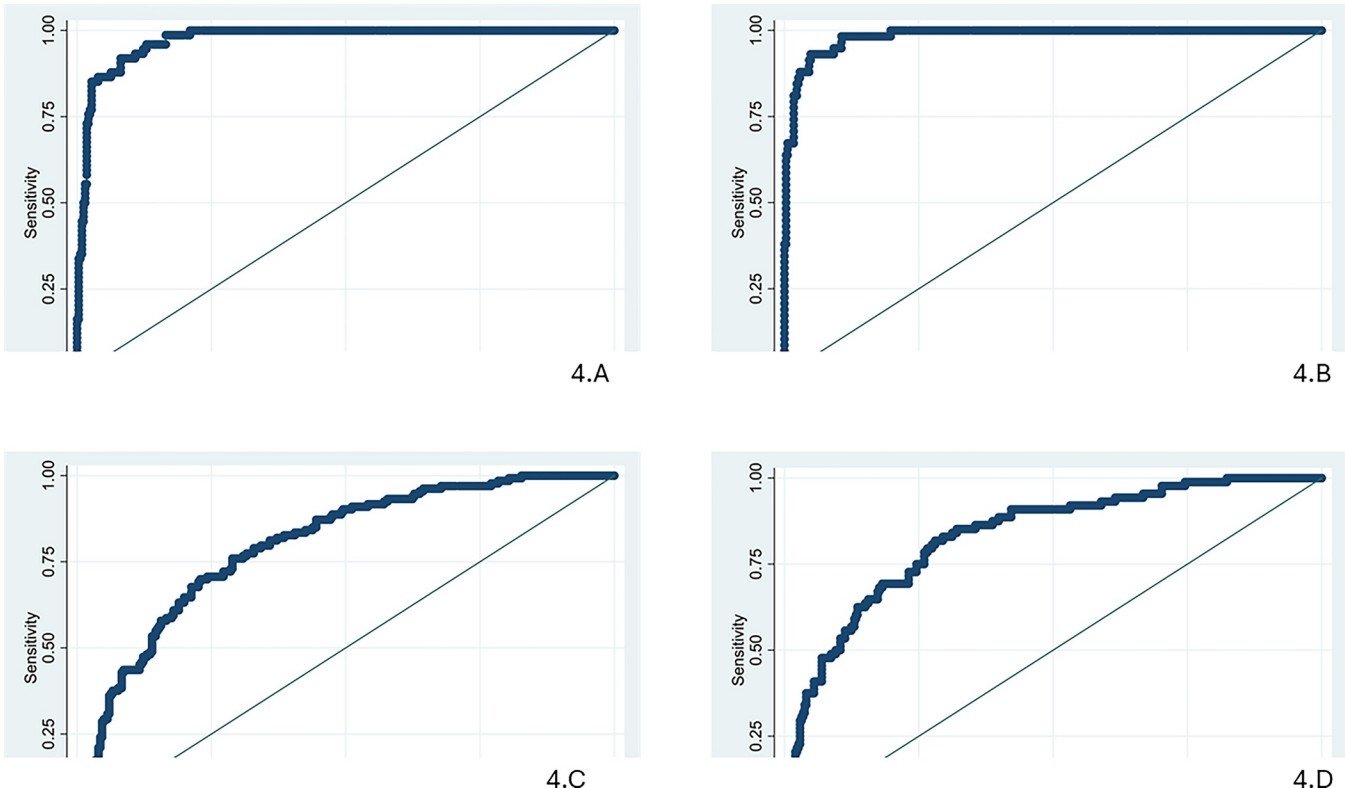

**Fig 4.** The ROC curve related to the model: A) curative service utilization for children aged 6 years and younger, B) preventive/consultation service utilization for children aged 6 years old and younger, C) curative service utilization for 7 to 12-year-old children, D) preventive/consultation service utilization for 7-to-12-year-old children.

aged 1–10 years was reported as 54.5% and 59.1% in West Virginia and Pennsylvania, respectively [20]. These frequencies are comparable to our findings. In contrast, a study in Al-Madinah, Saudi Arabia, showed a higher utilization rate of 76.2% among children aged 9–12 years old, possibly due to free access to dental services in the country [5]. A study on Canadian children indicated that 48.7% of children aged 1 to 4 and 90.3% of children aged 5 to 11 used preventive dental services in the past year, reflecting the publicly funded dental care for children in Canada [21].

## Predisposing factors

As per the findings of our study, the utilization of dental services shows a significant increase with age among children aged six years and younger. This trend, supported by other evidence [3,22,23], can be attributed to the accumulation of oral health issues over time, leading to higher utilization rates as children grow older. Moreover, younger children may need more communication skills, making it easier for parents to accurately perceive their dental needs as they are older [24]. Our study also revealed a positive association between age and parent-perceived oral health needs among children aged six years and younger, indicating that as children grew older, parents perceived more significant oral health needs and poorer oral health [22]. However, this relationship was different in children aged 7 to 12; as age increased, there was a decrease in parent-perceived oral health needs, and no significant association was found between age and parent-perceived oral health [25]. This finding explains the lack of a significant relationship between age and dental services utilization in this older age group.

A critical reason for delaying dental visits is the need for a better understanding of the importance of early dental care [25]. Previous studies have also underscored the role of knowledge in influencing dental services utilization [4,26]. A study in central Mexico showed that although knowledge significantly affected curative dental service utilization, its association with preventive dental service utilization was insignificant in adolescents [19]. Like this study, our study showed the partial effect of knowledge on dental service utilization. A higher knowledge score increased the odds of utilizing preventive/consultation services in children aged six and younger and curative services utilization in children aged 7 to 12. One reason for this finding could be that few children receive curative services in younger children and preventive services in older ones.

Dental visits before the age of one increased the odds of preventive/consultation dental services utilization in both age groups. A systematic review by Bhaskar et al. indicated that dental visits before age one were associated with more preventive visits and fewer curative dental treatments [27]. The increased odds of curative services utilization observed in our study among those who had dental visits before the age of one may reflect improved parental attitudes towards oral health and preserving children's teeth. Parental attitudes towards oral health are closely linked to the frequency of dental treatments received [28].

Oral health behavior significantly influenced preventive/consultation services utilization in children aged 7 to 12. A systematic review by Curi et al. highlighted that oral health behaviors such as tooth brushing and diet are critical determinants of dental services utilization, particularly preventive services in children aged 1 to 15 [3]. In our study, this association was insignificant in the younger age group. It is worth noting that the frequency of tooth brushing and snack consumption in this age group was lower compared to older children.

In our study, dental services utilization was similar in the two genders, so we did not add this variable in the final model. A study in India showed boys were more likely to use dental services in the past year [29]. By contrast, as a result of Aghili et al. in Saudia Arabia and Medina-Solis et al. in Nicaragua, dental service utilization was more distributed among girls [5,30]. The results regarding the effect of gender on dental service utilization were inconsistent and should be interpreted consciously. Different oral health statuses in two genders could be the reason for different dental service utilization. A meta-analysis in Iran found no difference between girls and boys in the DMFT/dmft index [31].

Another predisposing variable was the household head's education level, which positively correlated with socioeconomic status. Higher education provides better job opportunities, enhancing socioeconomic status [3]. This finding is consistent with a study by Gao et al., which highlighted the association between poverty and education [4].

## Enabling factors

According to our model, enabling factors not only directly influenced dental services utilization but also corresponded to individual needs.

Among these factors, better socioeconomic status was associated with increased odds of utilizing preventive/consultation services in both age groups and curative services in children aged 7 to 12. Socioeconomic status encompasses cultural and economic factors that shape parents' perceptions of oral health importance and behaviors, including dental service utilization, which aligns with Bourdieu's sociological theory [32]. Additionally, socioeconomic status broadens the choices for service delivery and subsequently enhances access [24].

This finding aligns with a study in Brazil that indicated children's economic status was associated with preventive and problem-based dental visits [33]. In our study, although not statistically significant, the odds of curative services utilization increased with improving

socioeconomic status among children aged six and younger. The cost of treatment is lower in early life, which may mitigate the influence of socioeconomic status in this age group [34].

Our results showed that lower socioeconomic status was associated with higher odds of parent-perceived oral health needs and poorer parent-perceived oral health. A previous study also found that children from low-income families had more oral health needs and lower dental services utilization [35].

Dental insurance, another enabling factor, showed no significant associations with the need component. Additionally, it was not related to dental services utilization except for curative services in children aged 7 to 12. Some studies have confirmed the role of insurance [3,36], whereas others have found no significant association [24,37]. Factors such as the type of insurance (private or public), characteristics of insured groups, and the nature of services provided may deter the role of insurance in dental services utilization [24,38,39].

Enabling factors predict potential access to dental services and the prerequisite of realized use [10]. Socioeconomic status and insurance coverage are not the only determinants of enabling factors. According to the Andersen model, regular sources of care are also important [10] and should be considered in future studies.

## Need factors

Andersen argued that need is the most immediate predictor of dental services utilization [22]. This aligns with our findings, which demonstrated that parent-perceived oral health needs ware the strongest predictor of curative services utilization in both age groups and preventive/consultation services in children aged 7 to 12. Previous studies have also highlighted the significant role of need in dental services utilization, whether perceived or evaluated [4,7,24,40]. Indeed, the need motivates people to seek dental services [7].

Parent-perceived oral health was another aspect considered in the need component. However, in our model, we did not find a significant relationship between parent-perceived oral health and dental services utilization. This finding is consistent with a study conducted in Al-Madinah, Saudi Arabia [5], which showed that dental services utilization in children was predicted by dental pain but not significantly associated with self-perceived oral health.

This study was conducted via a phone survey, which may introduce limitations such as the inability to utilize visual cues to establish rapport or instances where respondents may not be prepared for the interview during the call. Implementing an effective communication process and designing a suitable framework for telephone interviews are crucial for ensuring efficient data collection using this method [11]. Moreover, this method could not encompass individuals with no access to phones. With the extensive coverage of the landline (8200300 lines) and the considerable frequency of using cell phones in Tehran (72.6%), this issue was less troublesome [41,42]. In addition to its limitations, this method offers several advantages. These include broader geographic coverage as conducted centrally, guaranteeing the anonymity and comfort of the interviewer and interviewee and facilitating faster data collection [43,44]. Given these advantages, the challenges posed by the recent peak of the COVID-19 pandemic, the lower cost of telephone-based methods, and the significantly lower response rate of face-to-face methods in our society, we opted to employ this method for data collection in our study.

Another study limitation was that the information was parent-reported, which could lead to recall bias. We try to mitigate this limitation using standard questions from previous studies and another standard questionnaire.

The third limitation is the study's cross-sectional design, which cannot prove the causality relationship between explanatory variables and dental service utilization. Nevertheless, using a proper sample size and structural equation modeling based on the Andersen model, our study

has provided valuable results about the factors associated with preventive/consultation and curative services utilization separately across two age groups in all districts of Tehran. This categorization enabled us to discern the factors linked to these two types of dental care among age groups with distinct characteristics. Additionally, we employed a generalized structural equation model to construct a network illustrating the relationships among predictor variables, outcomes, and each other concurrently. These findings provide insights that can assist policymakers in targeting effective and modifiable determinants tailored to specific target populations.

## Conclusion

Among children aged 7–12, enabling factors such as socioeconomic status and dental insurance, along with need factors such as parent-perceived oral health need, played a crucial role in dental services utilization. In contrast, predisposing factors such as age significantly contributed to dental care utilization for younger children. Need factors emerged as strong predictors of dental services utilization. Policymakers should prioritize investigating modifiable factors associated with dental care utilization within each age group. Addressing these factors can enhance healthy behaviors and promote oral health across the population.

## Supporting information

**S1 File. Questionnaire for utilization of oral health services based on Andersen's behavioral model.**
(DOCX)

**S2 File. Data of study.**
(SAV)

## Author Contributions

**Conceptualization:** Elaheh Amirian, Samaneh Razeghi, Ahmad R. Shamshiri, Simin Z. Mohebbi.

**Data curation:** Elaheh Amirian, Ahmad R. Shamshiri, Simin Z. Mohebbi.

**Formal analysis:** Elaheh Amirian.

**Investigation:** Elaheh Amirian, Alireza Molaei, Ahmad R. Shamshiri, Simin Z. Mohebbi.

**Methodology:** Elaheh Amirian, Samaneh Razeghi, Alireza Molaei, Ahmad R. Shamshiri, Simin Z. Mohebbi.

**Project administration:** Simin Z. Mohebbi.

**Software:** Elaheh Amirian, Alireza Molaei, Ahmad R. Shamshiri.

**Supervision:** Samaneh Razeghi, Ahmad R. Shamshiri, Simin Z. Mohebbi.

**Validation:** Elaheh Amirian, Samaneh Razeghi, Alireza Molaei, Ahmad R. Shamshiri, Simin Z. Mohebbi.

**Writing – original draft:** Elaheh Amirian, Samaneh Razeghi, Alireza Molaei, Ahmad R. Shamshiri, Simin Z. Mohebbi.

**Writing – review & editing:** Elaheh Amirian, Samaneh Razeghi, Alireza Molaei, Ahmad R. Shamshiri, Simin Z. Mohebbi.

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
