## [Decision Letter · Decision Letter 0]

6 Aug 2024

PONE-D-24-26801Preventive and curative dental services utilization among children aged 12 years and younger in Tehran, Iran based on the Andersen behavioral model: A generalized structural equation modeling.PLOS ONE

Dear Dr. Mohebbi,

Thank you for submitting your manuscript to PLOS ONE. After careful consideration, we feel that it has merit but does not fully meet PLOS ONE’s publication criteria as it currently stands. Therefore, we invite you to submit a revised version of the manuscript that addresses the points raised during the review process.

We look forward to receiving your revised manuscript.

Kind regards,

Isabel Cristina Gonçalves Leite

Academic Editor

PLOS ONE

Journal Requirements:

2. In the ethics statement in the Methods, you have specified that verbal consent was obtained. Please provide additional details regarding how this consent was documented and witnessed, and state whether this was approved by the IRB.

3. We are unable to open your Supporting Information file "final data.sav". Please kindly revise as necessary and re-upload.

5. Please upload a new copy of Figures 2A, 2B and 3 as the detail is not clear. Please follow the link for more information: " ext-link-type="uri" xlink:type="simple">https://blogs.plos.org/plos/2019/06/looking-good-tips-for-creating-your-plos-figures-graphics/"
" ext-link-type="uri" xlink:type="simple">https://blogs.plos.org/plos/2019/06/looking-good-tips-for-creating-your-plos-figures-graphics/"

**Additional Editor Comments:**

There are considerations regarding the submitted manuscript for necessary adjustments.

Reviewers' comments:

Reviewer's Responses to Questions

**Comments to the Author**

1. Is the manuscript technically sound, and do the data support the conclusions?

Reviewer #1: Yes

Reviewer #2: Partly

2. Has the statistical analysis been performed appropriately and rigorously? 

Reviewer #1: Yes

Reviewer #2: No

3. Have the authors made all data underlying the findings in their manuscript fully available?

Reviewer #1: Yes

Reviewer #2: Yes

4. Is the manuscript presented in an intelligible fashion and written in standard English?

Reviewer #1: Yes

Reviewer #2: No

5. Review Comments to the Author

**Reviewer #1: **the study has been well conducted addressing the health care delivery system for Tehran child population, Kindly clarify and include following-

1. technique of sampling and randomization

2. calibration of 12 trainers

3. was there any parent who did not permit to participate or 886 parents were only contacted and 100% response rate was received. Kindly mention

4. Gender based health care utilization needs must be elaborated and discussed in Tehran

**Reviewer #2:** I appreciate the opportunity to evaluate this article. The article was skillfully crafted and well elucidated by the authors. Nevertheless, I have some questions regarding the manuscript. My observations are provided below.

Overall

• Grammatical mistakes are evident throughout the manuscript.

• Some parts appeared to have been plagiarized from multiple sources. Appropriate references and writing patterns must be modified.

Prominent source:

https://doi.org/10.1186/s12903-019-0996-x

https://doi.org/10.3390/ijerph18052491

Abstract section

This section is nicely written. I recommend that only the grammatical and abbreviation corrections be made.

Introduction section

The paragraph commences with an overview of the SDGs and UHC before transitioning to more specific objectives. Nevertheless, the transition could be more seamless to connect these concepts more effectively and employ more consistent terminology and transitions between them. The Andersen behavior model's introduction appears abrupt and inadequate, and it could be more effectively integrated with the antecedent content including appropriate referencing. The paragraph is somewhat disorganized due to the absence of a distinct topic sentence delineating the primary focus. It is necessary to expressly link this to the importance of dental services in pursuing the SDGs, including the inclusion of poorly described justification and overlapping similar information about the utilization of dental services. A more precise distinction, integration, and transition of these frameworks are recommended, with appropriate referencing and precise justification.

Methods section

The entire section is poorly written and lacks pertinent information. These sections require significant modifications in their written format. What is the rationale behind conducting telephonic interviews even in a non-pandemic era, while a face-to-face interview would be more appropriate? Details regarding the sampling method and sample selection must be incorporated, at least with minimal information. Were all the scales have been validated before use? If so, please provide details with the corresponding references. What was the average time to complete one interview? Did you adhere to any scoring guidelines for the questionnaires? If so, please provide a concise explanation with references.

Result section

This section is moderately well-described. Relatively minimal revisions are recommended. Poor comprehension and readability are the result of particular terminology consistency. Numbers and percentage values were mismatched in some variables with the explanation. So, a cross-check is recommended. The tables and figures could be better designed. Minor adjustments in table formatting are necessary. Additionally, figures 1 and 2 should be improved, as they appear to be haphazard.

Discussion and Conclusion section

The paragraph provides detailed statistical results but needs more clarity in presenting a comparative analysis. However, it must consistently elucidate the factors influencing utilization that these comparisons disclose. The significance of these comparisons and any observed trends or patterns must be more explicitly discussed. When referencing other studies, the text may require additional information regarding the context or connections to the current research. The discussion on predisposing factors could be more explicit in explaining the reasons for the presence or absence of specific associations. The enabling factors must thoroughly address the potential healthcare access limitations. The section on need factors necessitates distinguishing these discrepancies and exploring potential explanations. Moreover, it fails to provide sufficient information regarding implementing this telephonic survey method, the sampling method, the response rate, or any specific strategies to mitigate potential biases or limitations. The strengths, limitations, and recommendation sections should be explained more precisely.

Suggestion

The study questionnaire could be attached to the manuscript as a supplementary.

6. PLOS authors have the option to publish the peer review history of their article (what does this mean?). If published, this will include your full peer review and any attached files.

Reviewer #1: No

Reviewer #2: No

---

## [Author Response · Author response to Decision Letter 0]

28 Sep 2024

Journal requirement:

1- Please ensure that your manuscript meets PLOS ONE's style requirements, including those for file naming. we have now rechecked the instructions and addressed every point. 

2- In the ethics statement in the Methods, you have specified that verbal consent was obtained. Please provide additional details regarding how this consent was documented and witnessed, and state whether this was approved by the IRB. We considered the following points, which have now been added to the Material and Methods/Ethical consideration section. At the outset, the interviewers introduced themselves, clarified the research objectives, and assured participants were interested in being involved in the study. Participants were informed of their right to cease participation or withdraw from the study at any time. The interviewer asked the parent, “Do you wish to participate in this study? All interviews were recorded. We followed the methods of the California Health Interview Survey (CHIS). Please see Zlotnick C, Tam T, Ye Y. Statewide Policy Change in Pediatric Dental Care, and the Impact on Pediatric Dental and Physician Visits. Matern Child Health J. 2017 Oct;21(10):1939-1948.

3- We are unable to open your Supporting Information file "final data.sav". Please kindly revise as necessary and re-upload. 

The data file is now checked and re-uploaded.

4- When completing the data availability statement of the submission form, you indicated that you will make your data available on acceptance. We strongly recommend all authors decide on a data sharing plan before acceptance, as the process can be lengthy and hold up publication timelines. Please note that, though access restrictions are acceptable now, your entire data will need to be made freely accessible if your manuscript is accepted for publication. This policy applies to all data except where public deposition would breach compliance with the protocol approved by your research ethics board. If you are unable to adhere to our open data policy, please kindly revise your statement to explain your reasoning and we will seek the editor's input on an exemption. Please be assured that, once you have provided your new statement, the assessment of your exemption will not hold up the peer review process.

This has now been revised. The data is available and uploaded

5- Please upload a new copy of Figures 2A, 2B and 3 as the detail is not clear. We addressed this comment and uploaded higher quality figures 

6- Please include captions for your Supporting Information files at the end of your manuscript, and update any in-text citations to match accordingly.

We addressed this comment

Reviewer 1:

1. technique of sampling and randomization. More details were added in materials and method/ sample selection and data collection section, page 7 line 112.

2. calibration of 12 trainers: This has now been more clarified it in the methods and materials/ sample selection and data collection section, page 7, line 117.

3. was there any parent who did not permit to participate or 886 parents were only contacted and 100% response rate was received. Kindly mention. We have now clarified that in result/description of study population sections, page 12.

4. Gender based health care utilization needs must be elaborated and discussed in Tehran. Dental service utilization had a similar pattern in the two gender groups. we have now described it in Results/description of study population/predisposing factors and dental service utilization section, page 15 , and Discussion/ predisposing section, page 26.

Reviewer 2:

Grammatical mistakes are evident throughout the manuscript.

We have rechecked and solve the mistakes. Moreover, we have used language check service and attached the language check certificate.

Some parts appeared to have been plagiarized from multiple sources.

We have rechecked and solve the issue.

Introduction:

The paragraph commences with an overview of the SDGs and UHC before transitioning to more specific objectives. Nevertheless, the transition could be more seamless to connect these concepts more effectively and employ more consistent terminology and transitions between them. The Andersen behavior model's introduction appears abrupt and inadequate, and it could be more effectively integrated with the antecedent content including appropriate referencing. The paragraph is somewhat disorganized due to the absence of a distinct topic sentence delineating the primary focus. It is necessary to expressly link this to the importance of dental services in pursuing the SDGs, including the inclusion of poorly described justification and overlapping similar information about the utilization of dental services. A more precise distinction, integration, and transition of these frameworks are recommended, with appropriate referencing and precise justification.

The introduction has now been revised partially to address the comment. 

Methods:

What is the rationale behind conducting telephonic interviews even in a non-pandemic era, while a face-to-face interview would be more appropriate? We added more details about it in the materials and method section. The data collection was one year after the pandemic's peak, and the fear of the disease existed in Iran. Moreover, the results of pilot studies showed that telephone-based data collection was cheaper, and the response rate was significantly higher. These issues are now discussed in the paper and a reference has now been added in this regard in Material and methods section, page 6 and discussion section, pages 28 and 26

Details regarding the sampling method and sample selection must be incorporated, at least with minimal information. 

We revised the section and added more details related to the data collection method in the materials and method/ sample selection and data collection section.

Were all the scales have been validated before use? If so, please provide details with the corresponding references. The validation of the questionnaire is now described in more details in the material and methods/ questionnaire section.

What was the average time to complete one interview? We added that in the materials and methods/ Sample selection and data collection section, page 7.

Did you adhere to any scoring guidelines for the questionnaires? If so, please provide a concise explanation with references. We used scoring for the knowledge variable, explained in the materials and methods/ data handling and statistical analysis section. The other information was asked of the parent via a checklist that was provided according to the previous standard questionnaire (we cited them in the materials and methods/ questionnaire section).

Results:

Poor comprehension and readability are the result of particular terminology consistency. Numbers and percentage values were mismatched in some variables with the explanation. So, a cross-check is recommended. Cross-check was done and mistakes were revised.

The tables and figures could be better designed. Minor adjustments in table formatting are necessary. Additionally, figures 1 and 2 should be improved, as they appear to be haphazard. We addressed this comment and uploaded higher quality figures

Discussion:

The paragraph provides detailed statistical results but needs more clarity in presenting a comparative analysis. However, it must consistently elucidate the factors influencing utilization that these comparisons disclose. The significance of these comparisons and any observed trends or patterns must be more explicitly discussed. When referencing other studies, the text may require additional information regarding the context or connections to the current research. The discussion on predisposing factors could be more explicit in explaining the reasons for the presence or absence of specific associations. The enabling factors must thoroughly address the potential healthcare access limitations. The section on need factors necessitates distinguishing these discrepancies and exploring potential explanations. We considered your comment and added more explanation in the predisposing section and the role of enabling factors in potential access

Moreover, it fails to provide sufficient information regarding implementing this telephonic survey method, the sampling method, the response rate, or any specific strategies to mitigate potential biases or limitations. The strengths, limitations, and recommendation sections should be explained more precisely. 

We have now revised the discussion, limitations, and strengths to fulfill the comment. 

Suggestion:

The study questionnaire could be attached to the manuscript as a supplementary.

We have now uploaded the study questionnaire

---

## [Editor Report · Decision Letter 1]

30 Sep 2024

Preventive and curative dental services utilization among children aged 12 years and younger in Tehran, Iran based on the Andersen behavioral model: A generalized structural equation modeling.

PONE-D-24-26801R1

Dear Dr.Mohebi,

We’re pleased to inform you that your manuscript has been judged scientifically suitable for publication and will be formally accepted for publication once it meets all outstanding technical requirements.

Kind regards,

Isabel Cristina Gonçalves Leite

Academic Editor

PLOS ONE

---

## [Editor Report · Acceptance letter]

10 Dec 2024

PONE-D-24-26801R1 

PLOS ONE

Dear Dr. Mohebbi, 

I'm pleased to inform you that your manuscript has been deemed suitable for publication in PLOS ONE. Congratulations! Your manuscript is now being handed over to our production team.

Kind regards, 

on behalf of

Dr. Isabel Cristina Gonçalves Leite 

Academic Editor

PLOS ONE